# Survival outcome and prognostic factors of patients with nasopharyngeal cancer in Yogyakarta, Indonesia: A hospital-based retrospective study

Susanna Hilda Hutajulu[1]*, Daniel Howdon[2], Kartika Widayati Taroeno-Hariadi[1], Mardiah Suci Hardianti[1], Ibnu Purwanto[1], Sagung Rai Indrasari[3], Camelia Herdini[3], Bambang Hariwiyanto[3], Ahmad Ghozali[4], Henry Kusumo[5], Wigati Dhamiyati[5], Sri Retna Dwidanarti[5], I. Bing Tan[3,6], Johan Kurnianda[1], Matthew John Allsop[2]

1 Division of Hematology and Medical Oncology, Department of Internal Medicine, Faculty of Medicine, Public Health and Nursing, Universitas Gadjah Mada/ Dr Sardjito General Hospital, Yogyakarta, Indonesia, 2 Leeds Institute of Health Sciences, School of Medicine, Faculty of Medicine and Health, University of Leeds, Leeds, United Kingdom, 3 Department of Otorhinolaryngology, Head and Neck Surgery, Faculty of Medicine, Public Health and Nursing, Universitas Gadjah Mada/ Dr Sardjito General Hospital, Yogyakarta, Indonesia, 4 Department of Anatomical Pathology, Faculty of Medicine, Public Health and Nursing, Universitas Gadjah Mada/ Dr Sardjito General Hospital, Yogyakarta, Indonesia, 5 Department of Radiology, Faculty of Medicine, Public Health and Nursing, Universitas Gadjah Mada/ Dr Sardjito General Hospital, Yogyakarta, Indonesia, 6 Department of Otorhinolaryngology, Head and Neck Surgery, Maastricht University Medical Center, Maastricht, The Netherlands

* susanna.hutajulu@ugm.ac.id

Data Availability Statement: Data cannot be shared publicly because no regulation was applied

## Abstract

### Purpose

This study aimed to determine the survival outcome and prognostic factors of patients with nasopharyngeal cancer accessing treatment in Yogyakarta, Indonesia.

### Methods

Data on 759 patients with NPC diagnosed from 2007 to 2016 at Dr Sardjito General Hospital were included. Potential prognostic variables included sociodemographic, clinicopathology and treatment parameters. Multivariable analyses were implemented using semi-parametric Cox proportional hazards modelling and fully parametric survival analysis.

### Results

The median time of observation was 14.39 months. In the whole cohort the median observed survival was 31.08 months. In the univariable analysis, age, education status, insurance type, BMI, ECOG index, stage and treatment strategy had an impact on overall survival (OS) (p values <0.01). Semi-parametric multivariable analyses with stage stratification showed that education status, ECOG index, and treatment modality were independent prognostic factors for OS (p values <0.05). In the fully parametric models age, education status, ECOG index, stage, and treatment modality were independent prognostic factors for OS (p values <0.05). For both multivariable analyses, all treatment strategies were

on this issue in our local institution. Sharing of the full de-identified dataset is not possible due to restrictions imposed by the ethics committee as these are patient data, albeit de-identified, and it may be possible to determine the identify of participants given the extent of sociodemographic and clinical data available for each participant. Data are available from Ethics Committee (contact via email: mhrec_fmugm@ugm.ac.id) for researchers who meet the criteria for access to confidential data.

**Funding:** IBT received funding from The Dutch Cancer Society (KWF2012-5423). JK received funding from The Indonesian Society for Hematology and Medical Oncology Yogyakarta Branch. SHH received funding from Universitas Gadjah Mada's Publisher and Publication Board. We thank Jaap Middeldorp for having played a crucial role in acquiring the Dutch Cancer Society (KWF2012-5423) fund to enable the work to take place. The funders had no role in study design, data collection and analysis, decision to publish, or preparation of the manuscript.

**Competing interests:** The authors have declared that no competing interests exist.

associated with a reduced hazard (semi-parametric models, p values <0.05) and a better OS (parametric models, p values <0.05) compared with no treatment. Furthermore, compared with radiation alone or chemotherapy alone, a combination of chemotherapy and radiation either in a form of concurrent chemoradiotherapy (CCRT), sequential chemotherapy and radiation, or induction chemotherapy followed by CCRT demonstrated a reduced hazard (hazard ratio/HR 0.226, 95% confidence interval/CI 0.089–0.363, and HR 0.390, 95%CI 0.260–0.519) and a better OS (time ratio/TR 3.108, 95%CI 1.274–4.942 and TR 2.531, 95% CI 1.829–3.233) (p values < 0.01).

## Conclusions

Median OS for the cohort was low compared to those reported in both endemic and non-endemic regions. By combining the findings of multivariable analyses, we showed that age, education status, ECOG index, stage and first treatment modality were independent predictors for the OS.

## Introduction

Nasopharyngeal cancer (NPC) is a relatively uncommon malignancy associated with tobacco and alcohol use, and Epstein-Barr virus infection [1]. Worldwide, its annual incidence is 2.2/100,000 persons in males and 0.8/100,000 persons in females [2]. However, it is a major health problem in geographic areas including Southern China, Hong Kong, and South-East Asia [3–5]. In Indonesia, NPC is the third most prevalent cancer in males with age-standardized annual incidence rate of 10.5/100,000 males [6]. This rate is much higher than in Caucasian populations (0.5/100,000 per year in males) [7].

The Tumour, Node, Metastasis (TNM) classification system includes six groups of NPC based on tumor extension (T1-4), the involving nodals (N1-3), and the presence of metastasis (M0-1) [8]. Stages I-II are considered as early disease and stage III, IVA, and IVB as locoregional advance disease and Stage IVC as distant metastasis.

Radiotherapy remains the standard care for NPC treatment, either alone or in combination with chemotherapy [3, 4]. Radiation alone is generally applied in stage I disease. For stages II-IVB, guidelines recommend concurrent chemoradiotherapy (CCRT) with adjuvant chemotherapy (AC) (CCRT-AC), induction chemotherapy (IC) followed by CCRT (IC-CCRT), or CCRT alone [9, 10]. Combination chemotherapy and radiation treatment can also be applied sequentially [11, 12]. For stage IVC, platinum-based chemotherapy is the preferred strategy for chemonaive individuals with good performance [13], although chemoradiotherapy for locoregional disease can also be given [14].

Several studies have characterized differences in NPC survival in endemic areas. A higher probability for poor overall survival (OS) in more advanced stages has been reported in both endemic [15, 16] and non-endemic regions [17]. Histologically, risk of mortality is higher for WHO type I when compared to WHO types II and III [18], although other studies report contradictive findings [17]. A report from China in locally advanced cases treated with either CCRT or IC-CCRT showed a 5-year overall survival (OS) of 74.8% and 83.2%, respectively [19]. A study in Hong Kong demonstrated 5-year OS of 69.8% in patients diagnosed from 1976–2005. In Malaysia a 3-year OS rate was found at 94.3% for stage II, 80% for stage III and 79.8% for stage IV. The 3-year disease-free survival (DFS) rate was 90% for stage II, 80% for

stage III and 65% for stage IV [12]. Observations from non-endemic areas include a study in Slovenia that demonstrated a 5-year DFS of 59% and OS of 49.7% [20]. A Finnish study reported a cohort diagnosed between 1990 and 2009. This showed a 5-year OS of 57%, with OS for stages I-IV as 87%, 69%, 55% and 31%, respectively [17].

Beside disease stage, multiple factors such as sex and age have been identified to influence OS. Relative risk of death is lower for female and younger cases compared to male and older cases [4, 18, 19, 21], although some studies suggest younger age is associated with poorer DFS [22]. Studies also report education (duration >12 years), marital status (being married, divorced, or single, rather than widowed) and economic status (higher) as pre-treatment predictors of increased survival [10, 23, 24]. Furthermore, reports from the United States demonstrated that individuals with uninsured cases (where insurance can be considered an indicator of economic status) have a lower survival compared with cases possessing private insurance [25, 26].

A trend of mortality rate by date of diagnosis of NPC has been demonstrated in the research literature. An epidemiological study for the period 1980–1999 from the Hong Kong Cancer Registry outlined a steadily decreasing age-standardized incidence rate from 28.5 in 1980–84 to 20.2 in 1995–99 per 100,000 males, and from 11.2 to 7.8 per 100,000 females, resulting in a total decrease of 29% and 30%, respectively, over the 20-year period [27]. This was supported by more recent studies in other endemic areas [18, 28]. A study in Finland also demonstrated OS of patients treated in 1990–1999 at 49% and an OS of 63% for those treated in 2000–2009 [17].

The effect of pre-treatment nutritional status on survival of NPC patients remains contradictory. A study in China showed that underweight increased the risk of death and distant metastasis [29], whilst a study from Taiwan found no relationship between body mass index (BMI), weight loss and survival [30]. In regards to general performance, a Chinese study in patients with metachronous metastatic disease showed that Karnofsky performance index of <80 had a negative impact on the survival [31].

The variation of treatment modality may have an effect on survival. Globally, capabilities of radiotherapy technologies vary among centres, ranging from crude two-dimensional (2D) radiotherapy techniques to more sophisticated modalities, such as intensity-modulated radiation therapy (IMRT) [32]. Guidelines currently recommend IMRT as the preferred option [9, 10, 33]. A previous review of 1,593 consecutive cases treated with 2D radiotherapy, 3D radiotherapy and IMRT highlighted differing rates of 5-year DFS at 78%, 81% and 85%, respectively. The related 5-year OS of each modality was 71%, 73% and 80% [34]. A study in Malaysia assessed the long-term survival of 91 locally advanced NPC patients who were treated with conventional radical RT followed by adjuvant chemotherapy. The 5-year OS, DFS and loco-regional control rate was 80.1%, 76%, and 85%, respectively [12]. A study of 602 cases with stage IVA-B NPC treated with IMRT showed that the 5-year OS was 83.2% for IC-CCRT and 74.8% for CCRT alone [19]. Metastatic NPC cases are generally treated with palliative chemotherapy or added with radiation on loco-regional disease, with median survival ranges from 9.5–15 months [13, 14, 35].

Although NPC is highly prevalent in Indonesia, there is no comprehensive report on patients' survival with factors affecting survival not well understood or widely reported. The largest dataset to have been reported from Indonesia included an analysis of clinical and pathological features only [36, 37]. Indeed, there are few reports of survival analyses of NPC in Indonesia. The present study aims to address this gap in the literature by presenting a survival analysis of a patient population attending a teaching hospital in Yogyakarta, Indonesia, and explores prognostic parameters that have been reported in the literature. This work seeks to

develop the evidence base on factors that influence survival for NPC cases in Indonesia to guide the development of a suitable response in the provision of cancer care.

## Methods

### Study population

An NPC clinical registry was developed in 2012 at Dr Sardjito General Hospital Yogyakarta Indonesia. The registry includes data on all patients diagnosed and treated at the hospital from January 2007 to December 2016. Data were collected and collated from 2012 till 2018 and extracted from medical records and paired with data from the pathological department. Written informed consent was obtained from participants prior to medical treatment. This retrospective study that utilised data from the registry was approved by the Joint Ethics Committee of Faculty of Medicine, Public Health, and Nursing at Universitas Gadjah Mada and Dr Sardjito General Hospital (reference number: KE/FK/0250/EC/2018).

Details concerning patient characteristics, presentation of the disease, treatment and follow-up were obtained from medical records. The histological diagnosis was made according to the WHO classification of NPC [38]. Pre-treatment evaluations included gathering of sociodemographic characteristics (sex, age, educational status, marital status and insurance type), year of diagnosis, BMI, performance status (by the Eastern Cooperative Oncology Group/ECOG classification), a chest X-ray, computed tomography (CT) of the head-and-neck region, an abdominal ultrasound and a bone X-ray. CT scans of the abdominopelvic region or chest were conducted when clinically indicated. Patients underwent clinical staging according to the American Joint Committee on Cancer staging system. Staging was aligned with the 6th edition for diagnoses made prior to 2010 [39], and those from 2010 aligned with the 7th edition [8]. Staging was harmonised across levels of the 7th edition for the analysis.

### Treatment and follow-up

Treatment data was obtained from the clinical registry and ambulatory clinic charts. Generally, for adjuvant treatment intention, a conventional fractionated schedule of radiation (daily fraction of 2 Gy with a total 33–35 fractions) was given. In 2007–2016 2D and 3D conventional radiotherapy techniques were used. Early in 2017, application of IMRT was applied. Different strategies of chemotherapy for locally advanced disease were given due to the waiting time to radiation which included sequential chemotherapy and radiotherapy, CCRT alone, or IC-CCRT. For cases who had been planned to have sequential treatment or IC-CCRT, the TPF regimen (comprising of intravenous docetaxel 75 mg/m2 on day 1, cisplatin 100 mg/m2 on day 1 and 5-fluoro-uracil 1000 mg/m2/day on day 1–4, every 3 weeks) was applied. Alternatively, PF regimen (comprising of intravenous cisplatin 100 mg/m2/day on day 1 and 5-fluoro-uracil 1000 mg/m2/day on day 1–5, every 3 weeks) was applied for a 3–4 courses. In cases that needed to wait for long periods to begin a radiation schedule, chemotherapy courses with PF regimen were extended depending on the clinical conditions. In the CCRT scheme, concurrent chemotherapy consisted of intravenous cisplatin 40 mg/m2 weekly during radiotherapy. For patients with distant metastasis, palliative chemotherapy was the commonest treatment strategy applied using a PF regimen for 3 courses or other regimens depended on the clinical condition. Palliative bisphosphonate and radiotherapy was also applied on bone metastasis as needed.

Following treatment completion (either last cycle of chemotherapy or radiation), patients were followed up routinely with treatment response assessments planned between 8 and 12 weeks after treatment. This generally combined a physical assessment, nasoendoscopy, a nasopharyngeal CT-scan and other imaging for other part of the body. Following an initial follow-

up appointment, subsequent follow-up appointments were scheduled between 3 and 6 months during the first 2 years after all treatment completion. Follow-up data for clinical outcomes and patient survival were obtained from the medical records. Survival time is from cancer diagnosis until death or last follow-up. In addition, telephone communication was used for patients whose life status could not be found in the medical records. Family members of cases was considered as a respondent if the patient was deceased. Lost to follow up was determined when individual discontinued any cancer management or follow-up for any reason and life status was thus determined in the last hospital visit.

## Statistical analysis

We carried out statistical analysis using multivariable semiparametric Cox regression and multivariable parametric survival analysis. We estimated models both for a sample of the complete cases (i.e., where all variables are non-missing) and further imputed for the remaining cases with missing variables, giving a final imputed estimation sample of all cases with valid survival times. Cox regression imposes no assumptions about the underlying hazard function and estimates the mean multiplicative impact variable on the log-hazard of failure at any given time. This gives rise to estimated hazard ratios (HRs) which, if the proportional hazards (PH) assumption is not violated, can be interpreted as the impact of the variable on the instantaneous failure rate at any given time. If the PH assumption is violated, these HRs can still be interpreted as the geometric mean of such a relationship within the sample and, further, stratification for variables where such an assumption is violated can obviate any such problems. In our base case, we stratify by stage of NPC at entry to the study and further stratify in sensitivity analysis.

Fully parametric regression assumes an underlying baseline hazard of failure over time that can be specified as a function of time. Such models are estimated under either the PH assumption or the accelerated failure time (AFT) assumption. We estimate models for baseline hazards parameterised as exponential, lognormal, loglogistic, Gompertz, Weibull, and generalised gamma. We inform the choice of such models by comparing estimated Akaike Information Criteria (AIC) and Bayesian Information Criteria (BIC) metrics, and by examination of existing literature. Where missing values are exhibited for certain variables, we impute values under the assumption that these are missing at random. We further assume that right-censoring of our survival times due to loss to follow-up is non-informative.

## Results

We first present descriptive statistics and the Kaplan-Meier survival curves for our full sample. During this collection period, a total of 767 patients were registered. After exclusion of eight patients with missing or invalid survival times, we analysed a final 759 data out of 767 from the database. Characteristics of the study subjects are displayed in Table 1. The median age of subjects was 50 years. This cohort was similar in terms of age between cases under and over the median age. The majority of cases were male, completed education as recommended by the Indonesian government (a minimum of 9 years' study), were married, had government insurance (for civil servant, premium payer, or for the poor) or private insurance, were diagnosed before the introduction of national universal health coverage in 2014, had a low to normal BMI, with a good performance score, had locally advanced disease and treated firstly with chemotherapy or a combination of chemotherapy and radiation.

Patients were observed for a median time of 14.39 months. During the follow-up period, 266 cases (35%) died. A number of cases (n = 349, 46%) were considered as loss to follow-up and life status was drawn from the medical records. Although some patients are observed for

**Table 1. Characteristics of subjects (n = 759).**

| Characteristics | n | % |
|---|---|---|
| Age | | |
| <50 years | 388 | 51.1 |
| ≥50 years | 371 | 48.9 |
| Sex | | |
| Female | 218 | 28.7 |
| Male | 541 | 71.3 |
| Education (n = 649) | | |
| ≤9 years | 257 | 39.6 |
| >9 years | 392 | 60.4 |
| Marital status (n = 745) | | |
| Single | 58 | 7.8 |
| Married | 670 | 89.9 |
| Widowed | 17 | 2.3 |
| Insurance type (n = 723) | | |
| Government insurance for civil servant or premium payer or private | 300 | 41.5 |
| Self-financed | 113 | 15.6 |
| Government insurance for the poor | 310 | 42.9 |
| Year of diagnosis | | |
| 2014–2016 | 318 | 41.9 |
| Before 2014 | 441 | 58.1 |
| Body mass index (n = 708) | | |
| ≤22.3 | 499 | 70.5 |
| >22.3 | 209 | 29.5 |
| ECOG performance index (n = 501) | | |
| 0–1 | 361 | 72.1 |
| 2 | 126 | 25.2 |
| 3 | 14 | 2.8 |
| Stage at diagnosis (n = 750) | | |
| I-II | 53 | 7.1 |
| III-IVB | 581 | 77.5 |
| IVC | 116 | 15.5 |
| Treatment strategy (n = 643) | | |
| No treatment | 58 | 9.0 |
| Radiation only | 28 | 4.4 |
| Chemotherapy only | 171 | 26.6 |
| CCRT | 94 | 14.6 |
| Sequential chemotherapy and radiation | 192 | 29.9 |
| Chemotherapy-CCRT | 100 | 15.6 |

ECOG: Eastern Cooperative Oncology Group

CCRT: Concurrent chemotherapy and radiation

Sequential chemotherapy-radiation: chemotherapy followed by radiation or radiation followed by chemotherapy.

up to 145 months, we truncated the Kaplan Meier curve after the final death observed (at 46 months). The median survival time was 31.08 months (Fig 1).

We present results of the univariable analyses in Table 2 for various potential prognostic parameters. We found that age, education status, insurance type, BMI, performance status,

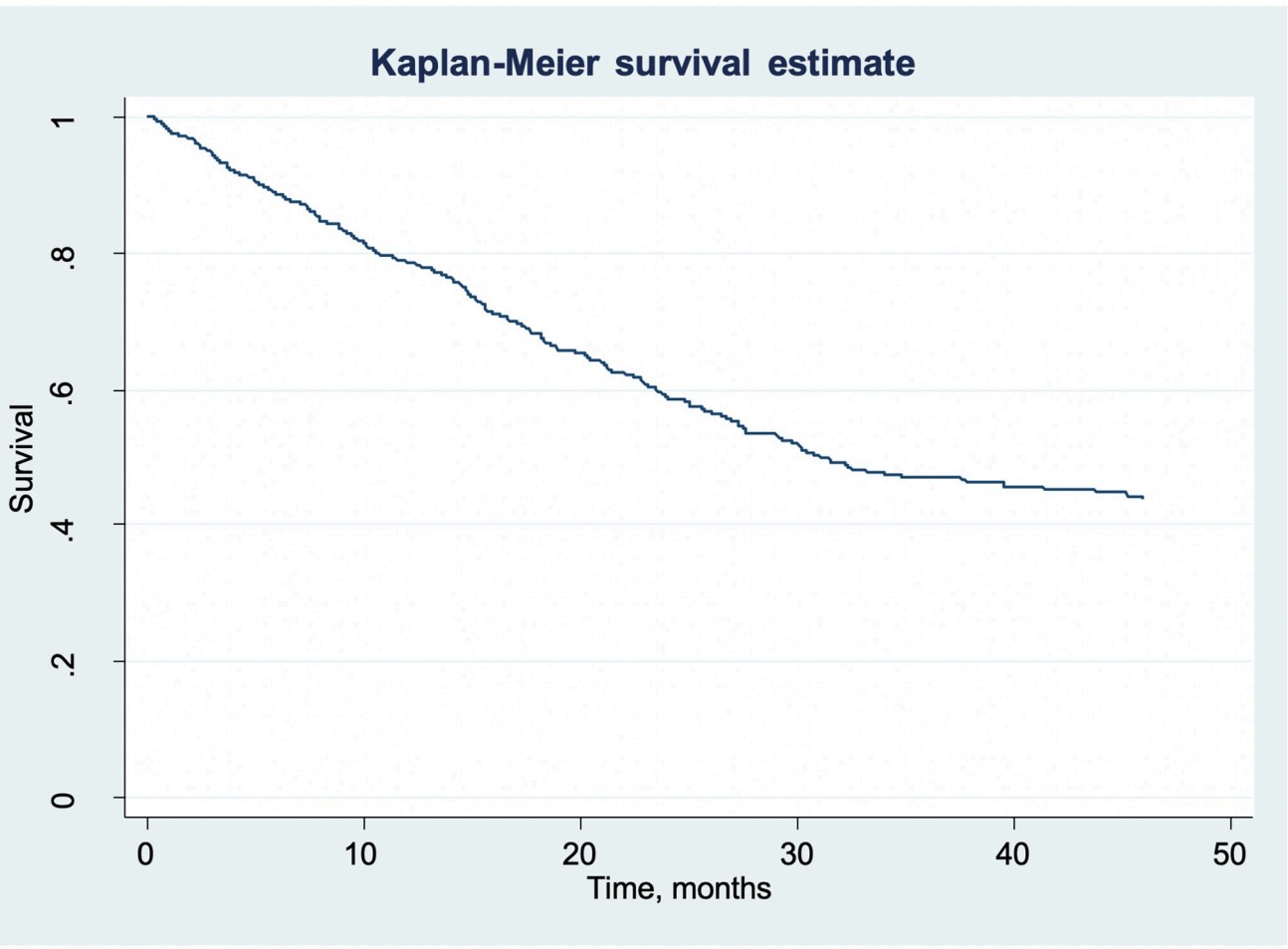

**Fig 1. Kaplan-Meier curve for the whole cohort.** Patients were observed for a median time of 14.39 months. The median survival time for the whole cohort was 31.08 months.

stage at entry and treatment strategy had impacted on the OS (p values< 0.01). We further estimated an initial set of Cox regressions under the proportional hazards assumption, with estimated HRs, significance levels of estimated coefficients, and 95% confidence intervals (CIs) for HRs presented. Although we only stratify in this base case by stage at entry (Table 3), results for the treatment group are robust to further stratification by all other variables included. This analysis showed that education status, ECOG index and first treatment modality were prognostic factors for OS (p< 0.01, p< 0.05 and p< 0.01). Individuals who had shorter education and poorer performance had a significantly reduced OS. Of note, HRs in all cases for these parameters are point-estimated at between 0.09 and 0.11. A hazard ratio of 0.1 implies that the hazard of failure for individuals with this characteristic have a reduction in the hazard of (1–0.1 = 90%). These results implied that, compared to no treatment, individuals receiving a sequential chemotherapy and radiation treatment or IC-CCRT had an approximately 90% reduction in the hazard of failure. Compared to radiation-only treatment, individuals receiving a combination of sequential chemotherapy and radiation had a reduced hazard point-estimated variously between 67% and 75%. Compared to radiation-only treatment, individuals receiving IC-CCRT had a reduced hazard point-estimated variously between 59% and

**Table 2. Univariable analyses for prognostic factors.**

| Characteristics | NPC cases (n = 759) | | | |
|---|---|---|---|---|
| | HR | | lower CI | upper CI |
| Age | | | | |
| <50 | Ref | | | |
| ≥50 | 1.703 | *** | 1.335 | 2.173 |
| Sex | | | | |
| Female | Ref | | | |
| Male | 1.068 | | 0.817 | 1.397 |
| Education | | | | |
| ≤9 years | Ref | | | |
| >9 years | 0.473 | *** | 0.362 | 0.618 |
| Marital status | | | | |
| Single | Ref | | | |
| Married | 1.287 | | 0.721 | 2.298 |
| Widowed | 1.499 | | 0.590 | 3.810 |
| Insurance type | | | | |
| Civil servant/premium payer/private | Ref | | | |
| Self-financed | 0.996 | | 0.662 | 1.500 |
| Government insurance for the poor | 1.688 | *** | 1.296 | 2.199 |
| Year of diagnosis | | | | |
| 2014–2016 | Ref | | | |
| Before 2014 | 0.951 | | 0.746 | 1.212 |
| Body mass index | | | | |
| ≤22.3 | Ref | | | |
| >22.3 | 0.605 | *** | 0.453 | 0.808 |
| ECOG performance index | | | | |
| 0–1 | Ref | | | |
| 2 | 2.044 | *** | 1.497 | 2.792 |
| 3 | 4.668 | *** | 2.277 | 9.571 |
| Stage at diagnosis | | | | |
| I-II | Ref | | | |
| III-IVB | 2.750 | *** | 1.410 | 5.364 |
| IVC | 4.694 | *** | 2.299 | 9.584 |
| Treatment strategy | | | | |
| No treatment | Ref | | | |
| Radiation only | 0.214 | *** | 0.120 | 0.383 |
| Chemotherapy only | 0.262 | *** | 0.183 | 0.377 |
| CCRT | 0.073 | *** | 0.045 | 0.120 |
| Sequential chemotherapy and radiation | 0.090 | *** | 0.061 | 0.131 |
| Chemotherapy-CCRT | 0.100 | *** | 0.065 | 0.155 |

HR: hazard ratio

CCRT: concurrent chemotherapy and radiation

*** $p < 0.01$.

**Table 3. Multivariable analyses using Cox regression (stratified by stage at diagnosis).**

| | Complete cases, n = 399 | | | | Multiple imputation, n = 759 | | | |
|---|---|---|---|---|---|---|---|---|
| | **HR** | | **lower CI** | **upper CI** | **HR** | | **lower CI** | **upper CI** |
| Age | | | | | | | | |
| <50 | 1 (ref) | | | | | | | |
| ≥50 | 1.352 | # | 0.970 | 1.884 | 1.259 | # | 0.967 | 1.638 |
| Sex | | | | | | | | |
| Female | 1 (ref) | | | | | | | |
| Male | 1.130 | | 0.788 | 1.622 | 0.936 | | 0.707 | 1.239 |
| Education status | | | | | | | | |
| ≤9 years | 1(ref) | | | | | | | |
| >9 years | 0.693 | # | 0.475 | 1.010 | 0.576 | *** | 0.418 | 0.795 |
| Marital status | | | | | | | | |
| Single | 1 (ref) | | | | | | | |
| Married | 2.269 | | 0.818 | 6.296 | 1.691 | # | 0.900 | 3.178 |
| Widowed | 2.296 | | 0.581 | 9.073 | 1.682 | | 0.624 | 4.534 |
| Insurance type | | | | | | | | |
| Civil servant/premium payer | 1 (ref) | | | | | | | |
| Self-financed | 1.013 | | 0.494 | 2.077 | 1.013 | | 0.624 | 1.643 |
| Insurance for the poor | 0.935 | | 0.630 | 1.388 | 0.938 | | 0.667 | 1.318 |
| Year of diagnosis | | | | | | | | |
| 2014–2016 | 1 (ref) | | | | | | | |
| Before 2014 | 1.305 | | 0.906 | 1.880 | 1.040 | | 0.766 | 1.411 |
| Body mass index | | | | | | | | |
| ≤22.3 | 1 (ref) | | | | | | | |
| >22.3 | 0.776 | | 0.536 | 1.125 | 0.795 | # | 0.581 | 1.087 |
| ECOG performance | | | | | | | | |
| 0–1 | 1 (ref) | | | | | | | |
| 2 | 1.523 | ** | 1.076 | 2.156 | 1.438 | ** | 1.042 | 1.984 |
| 3 | 2.607 | ** | 1.180 | 5.761 | 1.703 | | 0.719 | 4.037 |
| Treatment strategy | | | | | | | | |
| No treatment | 1 (ref) | | | | | | | |
| Radiation only | 0.432 | ** | 0.188 | 0.992 | 0.355 | *** | 0.186 | 0.680 |
| Chemotherapy only | 0.309 | *** | 0.170 | 0.560 | 0.263 | *** | 0.176 | 0.393 |
| CCRT | 0.065 | *** | 0.026 | 0.163 | 0.107 | *** | 0.060 | 0.191 |
| Seq. chemotherapy- radiation | 0.107 | *** | 0.057 | 0.199 | 0.108 | *** | 0.069 | 0.167 |
| Induction chemo-CCRT | 0.099 | *** | 0.047 | 0.206 | 0.106 | *** | 0.064 | 0.177 |

CCRT: concurrent chemotherapy and radiation

Seq. chemotherapy-radiation: chemotherapy followed by radiation or radiation followed by chemotherapy

# $p < 0.10$

** $p < 0.05$

*** $p < 0.01$.

77%. In addition, compared to radiation only and chemotherapy only, combination of chemotherapy and radiation either in CCRT, sequential scheme, or IC-CCRT had an association with better survival estimation (HR = 0.226, 95%CI 0.089–0.363, p = 0.001 and HR = 0.390, 95% CI 0.260–0.519, p< 0.01, respectively) (Table 4).

**Table 4. Treatment combination comparison.**

| Analysis | Treatment combination comparison | | | |
|---|---|---|---|---|
| | | **Hazard ratio** | **Lower CI** | **Upper CI** |
| **Semi-parametric regression** | CCRT+sequential chemo-radiation+chemo-CCRT vs radiation only | 0.226 *** | 0.089 | 0.363 |
| | CCRT+sequential chemo-radiation+ chemo-CCRT vs chemotherapy only | 0.390 *** | 0.260 | 0.519 |
| | | **Time ratio** | **Lower CI** | **Upper CI** |
| **Parametric regression** | CCRT+sequential chemo-radiation+chemo-CCRT vs radiation only | 3.108 *** | 1.274 | 4.942 |
| | CCRT+sequential chemo-radiation+chemo-CCRT vs chemotherapy only | 2.531 *** | 1.829 | 3.233 |

CCRT: concurrent chemotherapy-radiation

HR: hazard ratio (for semi-parametric analyses)

TR: time ratio (for fully parametric analyses)

*** p< 0.01.

We also estimated the above models employing parametric survival analysis (Table 5). For ease of precise interpretation of time ratios, we do not stratify our parametric analysis. We do, however, compare coefficients obtained in stratified analysis and results are robust to stratification. Further, the significance of staging coefficients in explaining ancillary parameters is not rejected at any conventional level (p-values estimated between 0.678 and 0.760). When comparing AIC and BIC metrics across our candidate baseline hazard functions, the best-performing is the lognormal. Further, existing literature suggests some support for the use of a lognormal parameterisation [40, 41]. Consequently, we estimate survival functions with this parameterisation. As lognormal survival analysis is parameterised under the AFT assumption, time ratios (indicating the multiplicative relationship with survival time) replace HRs in these results. Consequently, the interpretation of these coefficients is that a time ratio above 1 indicates a characteristic beneficial to survival, and a time ratio below 1 indicates a characteristic harmful to survival. Findings showed that individuals with older age, shorter education, poorer ECOG, and more advanced stage had a reduced OS compared to their counterparts. Individuals receiving treatment of any strategy had a better survival compared to individuals with no treatment. Combination of chemotherapy and radiation either in sequential scheme, CCRT, or IC-CCRT correlated with definitely better survival when compared to radiation-only or chemotherapy-only treatment (TR = 3.108, 95% CI 1.274–4.942, p< 0.01 and TR = 2.531, 95% CI 1.829–3.233, p< 0.01, respectively) (Table 4).

Fig 2a–2f demonstrated that cases with older age, shorter education, poorer ECOG status and more advanced disease had worse OS compared to their counterparts. Cases receiving no treatment, radiation only, and chemotherapy only also had worse survival compared to those receiving combination of chemotherapy and radiation either in sequential scheme, CCRT, or IC-CCRT. We also used estimates from our parametric model (as in Table 5) to estimate extrapolated survival beyond the maximum timeframe offered by our analysis, as shown in Fig 3. We estimated the OS at 5 years to be around 35%.

## Discussion

### Summary of key findings

This study presents the first comprehensive and most extensive report on the survival of NPC patients for Indonesian cases. Despite being an endemic region, data on survival of NPC cases in Indonesia has been underreported to date. In this cohort, patients' median OS of 31 months was relatively low compared to other reports from both endemic and non-endemic areas.

**Table 5. Multivariable analyses using parametric (lognormal) regression.**

| | Complete cases, n = 399 | | | | Multiple imputation, n = 759 | | | |
|---|---|---|---|---|---|---|---|---|
| | TR | | lower CI | upper CI | TR | | lower CI | upper CI |
| **Age** | | | | | | | | |
| <50 | 1 (ref) | | | | | | | |
| ≥50 | 0.810 | | 0.622 | 1.054 | 0.740 | ** | 0.584 | 0.937 |
| **Sex** | | | | | | | | |
| Female | 1 (ref) | | | | | | | |
| Male | 0.900 | | 0.681 | 1.190 | 1.067 | | 0.832 | 1.368 |
| **Education status** | | | | | | | | |
| ≤9 years | 1 (ref) | | | | | | | |
| >9 years | 1.368 | ** | 1.027 | 1.822 | 1.564 | *** | 1.185 | 2.064 |
| **Marital status** | | | | | | | | |
| Single | 1 (ref) | | | | | | | |
| Married | 0.559 | | 0.279 | 1.118 | 0.601 | # | 0.359 | 1.005 |
| Widowed | 0.604 | | 0.220 | 1.662 | 0.626 | | 0.259 | 1.512 |
| **Insurance type** | | | | | | | | |
| Civil servant/premium | 1 (ref) | | | | | | | |
| Self-financed | 0.833 | | 0.489 | 1.419 | 0.933 | | 0.614 | 1.419 |
| Insurance for the poor | 0.960 | | 0.705 | 1.309 | 0.961 | | 0.710 | 1.301 |
| **Year of diagnosis** | | | | | | | | |
| 2014–2016 | 1 (ref) | | | | | | | |
| Before 2014 | 1.007 | | 0.746 | 1.358 | 1.119 | | 0.852 | 1.470 |
| **Body mass index** | | | | | | | | |
| ≤22.3 | 1 (ref) | | | | | | | |
| >22.3 | 1.116 | | 0.839 | 1.484 | 1.143 | | 0.875 | 1.492 |
| **ECOG performance** | | | | | | | | |
| 0–1 | 1 (ref) | | | | | | | |
| 2 | 0.704 | ** | 0.530 | 0.935 | 0.710 | ** | 0.535 | 0.942 |
| 3 | 0.446 | ** | 0.216 | 0.924 | 0.621 | | 0.302 | 1.279 |
| **Stage** | | | | | | | | |
| I-II | 1 (ref) | | | | | | | |
| III-IVB | 0.374 | *** | 0.191 | 0.731 | 0.381 | *** | 0.218 | 0.667 |
| IVC | 0.298 | *** | 0.142 | 0.624 | 0.355 | *** | 0.189 | 0.668 |
| **Treatment strategy** | | | | | | | | |
| No treatment | 1 (ref) | | | | | | | |
| Radiation only | 2.484 | ** | 1.155 | 5.343 | 3.107 | *** | 1.621 | 5.958 |
| Chemotherapy only | 2.519 | *** | 1.468 | 4.323 | 3.492 | *** | 2.347 | 5.195 |
| CCRT | 8.439 | *** | 4.191 | 16.99 | 6.931 | *** | 4.235 | 11.343 |
| Seq. chemo-radiation | 6.777 | *** | 3.940 | 11.659 | 8.739 | *** | 5.774 | 13.227 |
| Induction chemo-CCRT | 6.738 | *** | 3.617 | 12.553 | 8.790 | *** | 5.527 | 13.981 |

CCRT: concurrent chemotherapy-radiation

Seq. chemo-radiation: sequential chemotherapy and radiation: chemotherapy followed by radiation or radiation followed by chemotherapy

# $p < 0.10$

** $p < 0.05$

*** $p < 0.01$.

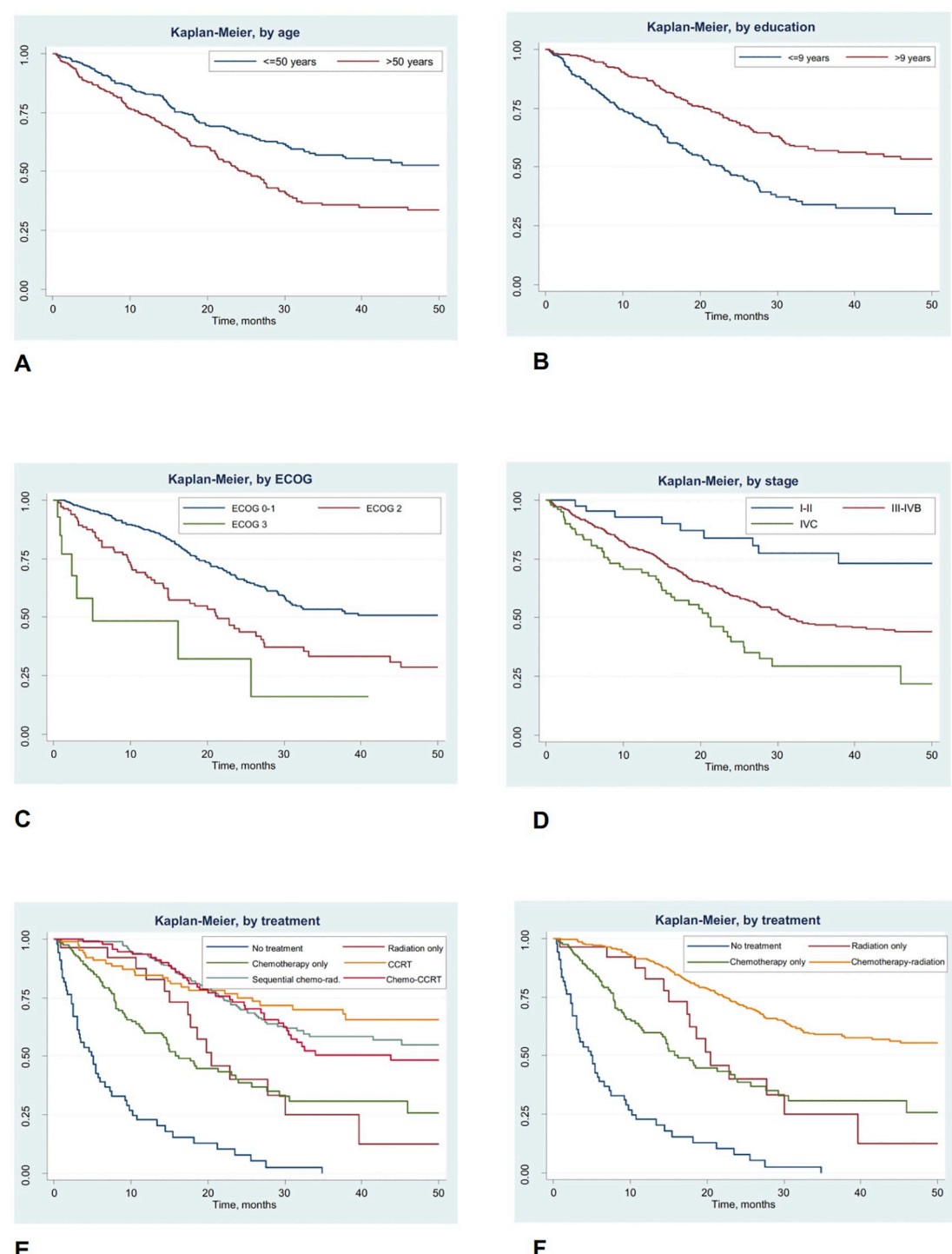

**Fig 2. Overall survival by multiple variables.** Individuals with older age (2a), shorter education (2b), poorer ECOG status (2c) and more advanced disease at entry (2d) had worse overall survival when compared to their counterparts. Individuals receiving no treatment, radiation only and chemotherapy only also had worse overall survival when compared to those receiving combination of chemotherapy and radiation either in sequential scheme, CCRT, or IC-CCRT (2e). Individuals receiving no treatment, radiation only and chemotherapy only definitely had worse overall survival when compared to those receiving combination of chemotherapy and radiation all together (2f).

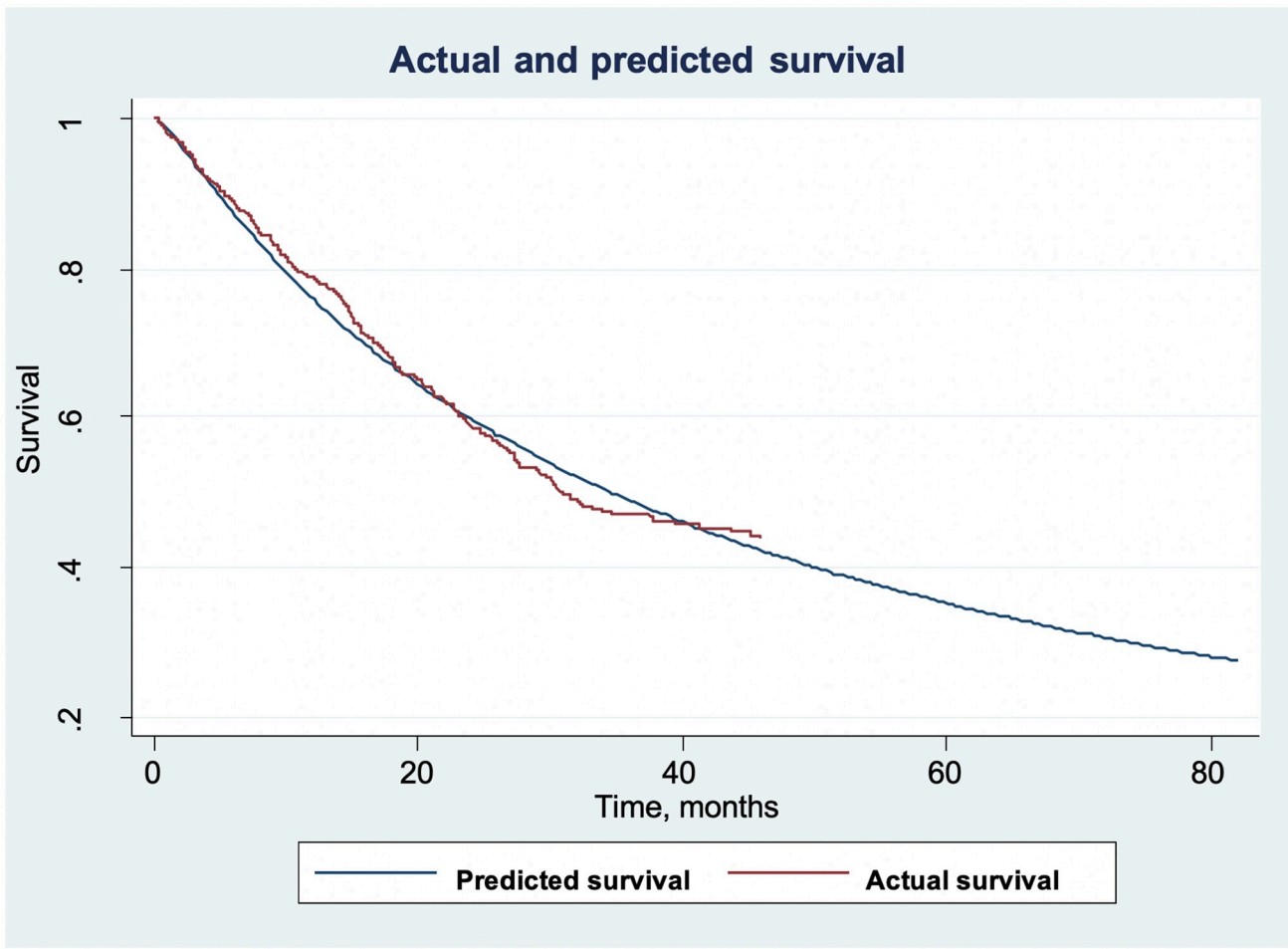

**Fig 3. Actual and predicted (and extrapolated) survival.** Estimates from parametric model (as shown in Table 5) was used to estimate extrapolated survival beyond the maximum timeframe offered by our analysis. The estimated 5-year overall survival is around 35%.

When extrapolated to 5-year survival time, the OS rate presented in this data only reached 35%. There were significant survival differences between cases with younger age, longer duration of education, better ECOG performance status, earlier clinical stage, and those receiving combination of chemotherapy and radiation, and their counterparts.

## Comparison with other studies in endemic and non-endemic areas

Low survival status of patients with NPC has been previously been reported from Indonesia. An earlier study by the team reported a 21-month median survival from 78 selected cases diagnosed from 2008–2011 which had a shorter follow-up time due to focusing on factors influencing the duration of radiation waiting lists [42]. Data from these cases were also included in the larger dataset reported in this manuscript. A further study in Indonesia demonstrated a 2-year OS of 39–71% in young, non-metastatic cases [37]. Combining this study with earlier reports, survival of Indonesian NPC patients is lower than those reported from surrounding endemic Asian countries which ranged from 54.7%-83.2 [12, 16, 19, 21, 43–45], and from non-endemic areas (ranging from 49.7%-69.1) for 5-year OS [17, 20, 46]. Studies that recruited only cases

with early disease demonstrated much better survival (94.5%-96.5%) [47, 48]. The predicted rate for 5-year survival for our present cohort (35%) matched only the survival rate of patients in the US four decades ago, between 1973–1979 [21].

## Prognostic factors and clinical and population health applicability

Prognostic factors favourable to survival in the present study included young age, education > 9 years, a good performance index, early stage of disease and undergoing a combination of chemotherapy and radiation treatment. Some studies reported various cut-off values (40–55 years) for univariable analyses regarding age [18–21, 28]. Our findings are consistent demonstrating a survival benefit in young age. Education has generally drawn attention as a crucial factor with worse survival in the lower educational groups [16, 49]. Large epidemiological studies have highlighted socioeconomic differences in cancer survival that are more pronounced than those in cancer incidence, with lower survival for individuals in disadvantaged socioeconomic conditions [50, 51]. However, teasing apart the causal argument from selection persists in making it difficult to determine the underlying mechanisms influencing this trend. The Indonesian government made it obligatory for an education duration of 9 years since 1994, 13 years prior to the intake of NPC cases in our current panel, but lower educational attainment persists. Alongside identifying and addressing barriers to completing minimum education years, there may be scope to explore health promotion approaches to encourage healthy lifestyles particularly in rural areas where education levels are reduced (see S1 Table).

Aligned with previous findings, performance index and nutritional status that was determined by BMI were found to be closely associated with stage of disease at presentation (see S2 Table) [31], with both generally being low in the presence of advanced disease [31]. Tumour site and stage may directly impair the oral intake of patients with cancer in the head and neck area and a low performance status may enlarge the risk of nutritional worsening [52]. Bozetti et al also confirmed an association of primary site and ECOG status with nutritional risk score [53]. However, nutritional status of our patients did not have a significant impact on their survival status.

The late presentation of NPC in the local setting of Yogyakarta needs to be addressed. At both the health service and patient level, known barriers to earlier presentation exist. At the health service level, a poor diagnosis by general practitioners in the local primary health centres has previously been found to lead to a delay of patient referral [54]. Similarly, findings in Malaysia highlighted the low knowledge and awareness of NPC by health professionals may have an impact on delayed diagnosis [55] and patient outcomes [56]. For patients, low awareness may be linked to lack of visibility of the nasopharyngeal region, so tumour development may not be evident initially. Additionally, NPC-associated symptoms often mimic manifestation of other chronic diseases in head and neck area, such as chronic rhinitis, hearing impairments and headache [36]. Denial of illness and economic constraints may become further factors influencing delayed diagnosis and treatment [57]. Once identified, national universal health coverage ensures access to necessary treatments for NPC patients in Indonesia following diagnosis. This study highlights increased survival when presenting and accessing such treatments at an earlier stage of NPC. Wider benefits to patients of increased access to medical treatments through national universal health coverage are starting to be identified [58]. However, efforts to increase early presentation must be met by simultaneous efforts to reduce convoluted referral processes that may produce long waiting times between presentation and diagnosis.

Treatment strategy was found to have an impact on the survival status in NPC cases in this study. This aligns with findings from previous literature. When appropriate to the stage and in the absence of problematic comorbidity, various treatment strategy for NPC can provide good clinical outcomes and favourable survival [12, 34, 59]. For all stage I NPC without very bulky tumours, receipt of radiation alone is recommended with best option using IMRT [10, 34]. The addition of platinum-based chemotherapy improves clinical outcomes in physically fit patients with stage II-IVB either in the form of IC-CCRT,CCRT-AC, or CCRT alone [14, 16, 34, 59–62]. Sequential chemotherapy and radiation can be another strategy, especially in stage II NPC [11]. In our local panel, patients received radiation from 2D and 3D machines except those treated from early 2017 when IMRT was firstly introduced. There was variation in clinical outcome and the best result was achieved through CCRT and CCRT-AC approaches. Sequential chemo-radiation was widely applied especially before 2014. This strategy often became an option considering the long waiting lists for radiation treatment as shown by two previous studies from our local cases [42, 63]. and to accommodate a lower toxicity in patients with low performance status. The findings of the study suggest that an optimal strategy for NPC patients is a combination of radiotherapy and chemotherapy in locally advanced patients. Combination can be in a form of CCRT and CCRT-AC, although sequential treatment could be considered for those with low ECOG and old age. Whilst recommendations of treatment strategies for NPC in Indonesia can be guided by this study, they need to account for a context with limited equipment for delivering radiation therapies and complex referral systems.

All prognostic parameters obtained from this study may serve as targets for improvement. While the Indonesian government pursues efforts to provide optimal equipment in all oncology centres, optimization and refinement of the existing health service delivery could be effective to give NPC patients better access to optimal treatments and which may positively influence survival. A key area to direct investment will be in radiotherapy equipment which is necessary to support increases in capacity to deliver optimal treatments for NPC, alongside support wider capacity in supporting treatment delivery for other cancer types.

## Strength and study limitation

A strength of this study is the inclusion of statistical analyses using fully parametric tests, supporting the first comprehensive report on Indonesian NPC survival. This also contributes to gaps in literature about NPC treatment in low- and middle-income countries. The study also has limitations. Our study design is retrospective, where it relies greatly on the availability of data from existing medical records, and may be subject to caveats when using routine data, such as potential coding errors by those creating and maintaining clinical records. A further limitation is the high number of patients lost to follow up in the life status (46%) which may introduce bias in the calculation of survival duration.

## Conclusion

This study provides a comprehensive report on the survival of NPC patients for Indonesian cases. Median OS for the cohort was low compared to those reported across international literature in both endemic and non-endemic regions. These findings highlight the need to develop strategies to improve the delivery of care and management of patients with NPC in areas such as Yogyakarta in Indonesia. Age, education, ECOG status, clinical stage and treatment strategy were independent predictors for the patients' OS and may serve as targets for future improvement initiatives.

## Supporting information

**S1 Table. Distribution of education status by living areas.**
(DOCX)

**S2 Table. Mean BMI grouped by NPC staging.**
(DOCX)

**S1 Data. Anonymized data set.**
(XLSX)

**S1 File.**
(PDF)

## Acknowledgments

The authors thank Wahyu Wulaningsih, Sindu Wisesa, Guntara Khuzairi and Norma Dewi Suryani for technical assistance.

## Author Contributions

**Conceptualization:** Susanna Hilda Hutajulu, I. Bing Tan, Johan Kurnianda.

**Data curation:** Susanna Hilda Hutajulu, Mardiah Suci Hardianti, Camelia Herdini, Ahmad Ghozali, Johan Kurnianda.

**Formal analysis:** Susanna Hilda Hutajulu, Daniel Howdon, Mardiah Suci Hardianti, Camelia Herdini, Matthew John Allsop.

**Funding acquisition:** Susanna Hilda Hutajulu, I. Bing Tan, Johan Kurnianda.

**Investigation:** Susanna Hilda Hutajulu, Kartika Widayati Taroeno-Hariadi, Mardiah Suci Hardianti, Ibnu Purwanto, Sagung Rai Indrasari, Camelia Herdini, Bambang Hariwiyanto, Ahmad Ghozali, Henry Kusumo, Wigati Dhamiyati, Sri Retna Dwidanarti, I. Bing Tan, Johan Kurnianda.

**Methodology:** Susanna Hilda Hutajulu, Daniel Howdon, Mardiah Suci Hardianti, Camelia Herdini, I. Bing Tan, Johan Kurnianda, Matthew John Allsop.

**Project administration:** Susanna Hilda Hutajulu, Mardiah Suci Hardianti.

**Resources:** Susanna Hilda Hutajulu, Kartika Widayati Taroeno-Hariadi, Camelia Herdini, Ahmad Ghozali, Henry Kusumo, Wigati Dhamiyati, Sri Retna Dwidanarti.

**Software:** Daniel Howdon, Mardiah Suci Hardianti, Matthew John Allsop.

**Supervision:** Susanna Hilda Hutajulu, Kartika Widayati Taroeno-Hariadi, Mardiah Suci Hardianti, Ibnu Purwanto, Sagung Rai Indrasari, Camelia Herdini, Henry Kusumo, Matthew John Allsop.

**Validation:** Susanna Hilda Hutajulu, Daniel Howdon, Mardiah Suci Hardianti, Bambang Hariwiyanto, Ahmad Ghozali, Wigati Dhamiyati, Matthew John Allsop.

**Visualization:** Matthew John Allsop.

**Writing – original draft:** Susanna Hilda Hutajulu, Daniel Howdon, Matthew John Allsop.

**Writing – review & editing:** Susanna Hilda Hutajulu, Daniel Howdon, Kartika Widayati Taroeno-Hariadi, Mardiah Suci Hardianti, Ibnu Purwanto, Sagung Rai Indrasari, Camelia

Herdini, Bambang Hariwiyanto, Ahmad Ghozali, Henry Kusumo, Wigati Dhamiyati, Sri Retna Dwidanarti, I. Bing Tan, Johan Kurnianda, Matthew John Allsop.

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
