## [Decision Letter · Decision Letter 0]

2 Oct 2020

PONE-D-20-11649

Survival outcome and prognostic factors of patients with nasopharyngeal cancer in Yogyakarta, Indonesia: a hospital-based retrospective study

PLOS ONE

Dear Dr. Hutajulu,

Thank you for submitting your manuscript to PLOS ONE. After careful consideration, we feel that it has merit but does not fully meet PLOS ONE’s publication criteria as it currently stands. Therefore, we invite you to submit a revised version of the manuscript that addresses the points raised during the review process.

We look forward to receiving your revised manuscript.

Kind regards,

Gayle E. Woloschak, PhD

Academic Editor

PLOS ONE

Journal Requirements:

3. We noted in your submission details that a portion of your manuscript may have been presented or published elsewhere:

'The abstract was modified and has been published in the proceeding of ESMO Asia 2019 congress by The Annals of Oncology (https://www.annalsofoncology.org/article/S0923-7534(19)57964-9/fulltext). We uploaded a copy of the abstract from The Annals of Oncology. The present submission does not equate to dual publication because only an excerpt of findings was presented as part of ESMO abstract. The comprehensive reporting including multiple unseen and previously unreported analyses are available in this current manuscript. '

Please clarify whether this conference proceeding or publication was peer-reviewed and formally published.

If this work was previously peer-reviewed and published, in the cover letter please provide the reason that this work does not constitute dual publication and should be included in the current manuscript.

Additional Editor Comments:

The two reviewers have recommended major concerns for the work as noted in their comments. Please address these in a revision.

Reviewers' comments:

Reviewer's Responses to Questions

**Comments to the Author**

1. Is the manuscript technically sound, and do the data support the conclusions?

Reviewer #1: Yes

Reviewer #2: Partly

2. Has the statistical analysis been performed appropriately and rigorously? 

Reviewer #1: Yes

Reviewer #2: No

3. Have the authors made all data underlying the findings in their manuscript fully available?

Reviewer #1: Yes

Reviewer #2: No

4. Is the manuscript presented in an intelligible fashion and written in standard English?

Reviewer #1: Yes

Reviewer #2: Yes

5. Review Comments to the Author

Reviewer #1: The manuscript reports results of a retrospective survival analysis of patients who attended a hospital in Yogyakarta, Indonesia, and explores prognostic models. The study methods are basically reasonable, and well presented. I have the following comments:

(1) Semiparametric models and parametric models are both studied. In the abstract, the author chooses to report univariate semiparametric models and the multivariable parametric model. I am wondering why not report semiparametric models for both univariate and multivariable analysis, because doing this way it looks more consistent, and the semiparametric models are more popular where the baseline hazard function don’t need to be specified and hence is not sensitive to mis-specification.

(2) The abstract states “Compared with chemotherapy or radiation alone, a combination of chemotherapy and radiation demonstrated a significant benefit to OS (TRs variously estimated between 1.985 and 2.829).” Why not report CI for this comparison. Also, I don’t see these numbers (i.e., 1.985, 2.829) reported in the results section.

(3) In Table 2, the p-values of age and education should be marked as significant.

(4) Figure 2 is a little too blurry, it is hard to see the words and numbers in the figure. Please revise.

(5) In Table 2, “bivariable analysis” should be “univariate analysis”.

Reviewer #2: Introduction:

Introduction is far too long and reads more like a discussion/review paper. Majority of this information can be edited out, included in the discussion section or simply have citations provided.

Lines 69-73: The TNM classificiation system has 6 groups for NPC based on tumor extension. I, II, III, IVA, IVB, IVC.

Methods:

Lines 169-170: Were all the sociodemographic characteristics identified in the medical records?

Line 160: Presenting the number of patients collected would be a result and should not be included in the methods section.

Results:

Why include metastatic patients in the analysis of, do you feel this is driving some of the prognostic factors?

Discussion:

Lines 434: The authors refer to treatment strategy was found to have an impact on survival status, it appears they are comparing no treatment to treatment?

6. PLOS authors have the option to publish the peer review history of their article (what does this mean?). If published, this will include your full peer review and any attached files.

Reviewer #1: No

Reviewer #2: No

---

## [Author Response · Author response to Decision Letter 0]

1 Dec 2020

Dear editor and reviewers,

We are thankful for the positive feedback received from reviewers 1 and 2, and the editorial team, and for the opportunity to respond to the constructive points in our accompanying revised manuscript. Please find below our point-by-point response to reviewer feedback outlining, where relevant, the related changes we have made. We have uploaded revised versions of the manuscript as instructed, including both a clean copy and a track changes version. 

Editor comments

Reply: We have checked these requirements ahead of submitting our revised manuscript. 

2. We note that you have indicated that data from this study are available upon request. PLOS only allows data to be available upon request if there are legal or ethical restrictions on sharing data publicly. For information on unacceptable data access restrictions.In your revised cover letter, please address the following prompts: a) If there are ethical or legal restrictions on sharing a de-identified data set, please explain them in detail (e.g., data contain potentially identifying or sensitive patient information) and who has imposed them (e.g., an ethics committee). Please also provide contact information for a data access committee, ethics committee, or other institutional body to which data requests may be sent. b) If there are no restrictions, please upload the minimal anonymized data set necessary to replicate your study findings as either Supporting Information files or to a stable, public repository and provide us with the relevant URLs, DOIs, or accession numbers. Please see http://www.bmj.com/content/340/bmj.c181.long for guidelines on how to de-identify and prepare clinical data for publication. For a list of acceptable repositories, please see http://journals.plos.org/plosone/s/data-availability#loc-recommended-repositories. We will update your Data Availability statement on your behalf to reflect the information you provide.

Reply: In our cover letter we have provided details about data shared to accompany our manuscript. We have provided a supplementary file which includes tables of all data used to generate the figures and analyses presented in the paper. Sharing of the full de-identified dataset is not possible due to restrictions imposed by the ethics committee as these are patient data, albeit de-identified, and it may be possible to determine the identify of participants given the extent of sociodemographic and clinical data available for each participant. Should there be a request for data, this can be sent to the corresponding author (email: susanna.hutajulu@ugm.ac.id). Future researchers can contact the institutional ethics committee (email: mhrec_fmugm@ugm.ac.id) at Universitas Gadjah Mada, Indonesia with data access queries as well.

3. We noted in your submission details that a portion of your manuscript may have been presented or published elsewhere:

'The abstract was modified and has been published in the proceeding of ESMO Asia 2019 congress by The Annals of Oncology (https://www.annalsofoncology.org/article/S0923-7534(19)57964-9/fulltext). We uploaded a copy of the abstract from The Annals of Oncology. The present submission does not equate to dual publication because only an excerpt of findings was presented as part of ESMO abstract. The comprehensive reporting including multiple unseen and previously unreported analyses are available in this current manuscript. '

Please clarify whether this conference proceeding or publication was peer-reviewed and formally published.

If this work was previously peer-reviewed and published, in the cover letter please provide the reason that this work does not constitute dual publication and should be included in the current manuscript.

Reply: We have added the following information to the cover letter: “The conference proceeding was not peer-reviewed. The abstract, submitted to the ESMO Asia conference 2019, was assessed for relevance and quality by an abstract review committee. The work was selected for oral presentation, reflecting a positive appraisal. However, the work presented in this manuscript covers a more extensive analysis and overview of the data, presents new findings, and all contextual information (introduction, methods, findings and discussion) that were not possible to include in the conference abstract.”

Reviewer 1

The manuscript reports results of a retrospective survival analysis of patients who attended a hospital in Yogyakarta, Indonesia, and explores prognostic models. The study methods are basically reasonable, and well presented.

Reply: Many thanks for your feedback and further comments which have helped us to refine the content of the manuscript.

1. Semiparametric models and parametric models are both studied. In the abstract, the author chooses to report univariate semiparametric models and the multivariable parametric model. I am wondering why not report semiparametric models for both univariate and multivariable analysis, because doing this way it looks more consistent, and the semiparametric models are more popular where the baseline hazard function don’t need to be specified and hence is not sensitive to mis-specification.

Reply: While Table 5 indeed presents multivariable parametric results, Table 3 presents multivariate regression output for the semiparametric Cox model. In line with your suggestion, we have also amended our abstract to report both the univariable and multivariable analyses. 

2. The abstract states “Compared with chemotherapy or radiation alone, a combination of chemotherapy and radiation demonstrated a significant benefit to OS (TRs variously estimated between 1.985 and 2.829).” Why not report CI for this comparison? Also, I don’t see these numbers (i.e., 1.985, 2.829) reported in the results section.

Reply: Many thanks for highlighting this data presented in the abstract. The data reported were taken from a supplementary file which accompanied the original submission. In line with your suggestions, we have included confidence intervals in the reporting of findings in the manuscript. We appreciate that this is an important finding, hence its inclusion in the abstract, and have moved the supporting data into a table in the main body of the manuscript to increase its accessibility to other readers. The data in abstract (i.e., 1.985, 2.829) referred to a previous superseded version which we have now amended. We have now ensured these data are clearly and consistently reported throughout the abstract, Table 4 and findings section in the main body of the manuscript.

3. In Table 2, the p-values of age and education should be marked as significant.

Reply: We are grateful to the reviewer for noticing this error. A systematic recheck has been made on significance levels for all tables and these two errors, which have been corrected. This error arose in the final drafting process after a change of how we systematically categorised significance levels. This has now been reverted to the fuller categorisation which indicates variable significant at 10%, 5% and 1% levels. We have opted to maintain the Asterix symbol for reflecting significance levels at 5% and 1% and have used a different symbol (i.e. #) to indicate significance at 10% so as not to cause confusion to readers.

4. Figure 2 is a little too blurry, it is hard to see the words and numbers in the figure. Please revise.

Reply: Thank you for highlighting this issue. We have formatted the figure file in line with journal requirements and have used the PACE platform to now generate a .tif version of the file. We have included the .tif file alongside the revised manuscript and, should the manuscript be accepted, can work with the typesetting team to ensure that the figure is displayed in its final format in a way that is easy to view by readers.

5. In Table 2, “bivariable analysis” should be “univariate analysis”.

Reply: We are grateful for the reviewer’s close reading on the paper on this point and have corrected our title for this table.

Reviewer 2

1. Introduction: Introduction is far too long and reads more like a discussion/review paper. Majority of this information can be edited out, included in the discussion section or simply have citations provided.

Reply: We thank you for this point. We were mindful that the reporting of NPC survival in Indonesia is very limited, so were keen to ensure the context of existing literature and cancer provision relating to NPC in Indonesia were both clear outlined. However, we appreciate that readers typically prefer succinct and concise writing. As such, we have reviewed the content of the introduction, reducing the word count where possible and ensured the writing is as succinct as possible whilst retaining important contextual information for framing the analysis, interpretation of findings and the discussion section.

2. Lines 69-73: The TNM classification system has 6 groups for NPC based on tumour extension. I, II, III, IVA, IVB, IVC.

Reply: Thank you for noting this error which has now been amended to reflect the six groups in the TNM classification system.

3. Methods: Lines 169-170: Were all the sociodemographic characteristics identified in the medical records?

Reply: Yes, to clarify, all the sociodemographic characteristics were identified in the medical records. We have amended the text in the methods section to reflect this, which now reads: ”Details concerning patient characteristics, presentation of the disease, treatment and follow-up were obtained from medical records”.

4. Methods: Line 160: Presenting the number of patients collected would be a result and should not be included in the methods section.

Reply: We have removed this sentence from the methods section and added it to the findings section.

5. Results: Why include metastatic patients in the analysis of, do you feel this is driving some of the prognostic factors?

Reply: The stage of disease is a widely known prognostic factor in all cancer types including nasopharyngeal cancer. However, there are mixed findings across survival analyses across clinical studies where, for example, stage III patients may have a worse prognosis than stage IV disease. Thus, we wanted to clarify the impact of the full spectrum of disease stage on case survival in our local population. As highlighted in the paper, advanced disease had worse overall survival compared to their counterparts. Furthermore, staging of disease was used to undertake stratification in the multivariable analyses as described in methods. 

6. Discussion: Lines 434: The authors refer to treatment strategy was found to have an impact on survival status, it appears they are comparing no treatment to treatment?

Reply: Thank you for this point. Rather than comparing no treatment to treatment, we explored the impact of a range of treatment strategies alongside no treatment. Treatment strategy was observed to have an impact on survival status in the study population. As outlined in Tables 2 - 5, we compared cases with no treatment (as reference) with different strategies (radiation only, chemotherapy only, CCRT, sequential chemoradiation, and IC-CCRT), stratified by stage at diagnosis for multivariable analyses using Cox regression. Naturally we hope that all treatment strategies would impact on favourable survival compared with no treatment and in this study, our findings would suggest this, both in semi-parametric and fully parametric analyses. With both analyses we also demonstrated that among cases with different treatment strategies, there were reduced hazards for patients receiving CCRT, sequential chemotherapy – radiation and IC-CCRT (Table 3) and better survival of those patients (Table 5). Further, in Table 4, added in this revised version, we also compared patients receiving combination of chemotherapy and radiation in a form of CCRT, sequential chemo-radiation or IC-CCRT to patients receiving radiation only and chemotherapy only, and observed that patients receiving combination had better survival compared with radiation and chemotherapy only. These are based on both semi-parametric and parametric analyses.

---

## [Decision Letter · Decision Letter 1]

25 Jan 2021

Survival outcome and prognostic factors of patients with nasopharyngeal cancer in Yogyakarta, Indonesia: a hospital-based retrospective study

PONE-D-20-11649R1

Dear Dr. Hutajulu:

We’re pleased to inform you that your manuscript has been judged scientifically suitable for publication and will be formally accepted for publication once it meets all outstanding technical requirements.

Kind regards,

Gayle E. Woloschak, PhD

Section Editor

PLOS ONE

Additional Editor Comments (optional):

Thank you for addressing concerns raised.

Reviewers' comments:

Reviewer's Responses to Questions

**Comments to the Author**

1. If the authors have adequately addressed your comments raised in a previous round of review and you feel that this manuscript is now acceptable for publication, you may indicate that here to bypass the “Comments to the Author” section, enter your conflict of interest statement in the “Confidential to Editor” section, and submit your "Accept" recommendation.

Reviewer #1: All comments have been addressed

Reviewer #2: (No Response)

2. Is the manuscript technically sound, and do the data support the conclusions?

Reviewer #1: Yes

Reviewer #2: Yes

3. Has the statistical analysis been performed appropriately and rigorously? 

Reviewer #1: Yes

Reviewer #2: Yes

4. Have the authors made all data underlying the findings in their manuscript fully available?

Reviewer #1: Yes

Reviewer #2: Yes

5. Is the manuscript presented in an intelligible fashion and written in standard English?

Reviewer #1: Yes

Reviewer #2: Yes

6. Review Comments to the Author

Reviewer #1: (No Response)

Reviewer #2: Would defer to editor if introduction length is still too long as it is currently 4 pages in length.

Minor error in introduction, repeats major health issue twice in same sentence after edit.

7. PLOS authors have the option to publish the peer review history of their article (what does this mean?). If published, this will include your full peer review and any attached files.

Reviewer #1: No

Reviewer #2: No

---

## [Editor Report · Acceptance letter]

2 Feb 2021

PONE-D-20-11649R1 

Survival outcome and prognostic factors of patients with nasopharyngeal cancer in Yogyakarta, Indonesia: a hospital-based retrospective study 

Dear Dr. Hutajulu:

I'm pleased to inform you that your manuscript has been deemed suitable for publication in PLOS ONE. Congratulations! Your manuscript is now with our production department. 

Kind regards, 

on behalf of

Dr. Gayle E. Woloschak 

Section Editor

PLOS ONE